# Thermodynamics of Quantum Spin-Bath Depolarization

**DOI:** 10.3390/e25020340

**Published:** 2023-02-13

**Authors:** Durga Bhaktavatsala Rao Dasari

**Affiliations:** 3. Physikalisches Institut, University of Stuttgart, 70569 Stuttgart, Germany; d.dasari@pi3.uni-stuttgart.de

**Keywords:** quantum thermodynamics, quantum information, quantum spin-baths

## Abstract

We analyze here through exact calculations the thermodynamical effects in depolarizing a quantum spin-bath initially at zero temperature through a quantum probe coupled to an infinite temperature bath by evaluating the heat and entropy changes. We show that the correlations induced in the bath during the depolarizing process does not allow for the entropy of the bath to increase towards its maximal limit. On the contrary, the energy deposited in the bath can be completely extracted in a finite time. We explore these findings through an exactly solvable central spin model, wherein a central spin-1/2 system is homogeneously coupled to a bath of identical spins. Further, we show that, upon destroying these unwanted correlations, we boost the rate of both energy extraction and entropy towards their limiting values. We envisage that these studies are relevant for quantum battery research wherein both charging and discharging processes are key to characterizing the battery performance.

## 1. Introduction

Solid-state spin defects have become a resource for many quantum information protocols and at the same time also a test-bed for demonstrating quantum effects all the way to room temperature [1,2]. A key aspect in using these systems for quantum technologies is the hyperpolarization wherein a readily polarizable central (probe) spin transfers to and polarizes a quantum nuclear-spin register made of qubits with very long coherence times [3,4]. Further applications of these systems as memory and energy storage units [5] would also require optimal strategies for the extraction of stored energy (depolarization) or retrieve the quantum states stored during a quantum communication protocol [6]. We will focus here on the depolarization process with a focus on its thermodynamical applications.

In hyperpolarization schemes, one resonantly transfers polarization from an electron spin to a nuclear spin via the well-known Hartmann Hahn (HH) interaction [7]. This transfer is complete for a single spin and for a bath of non-identical spins wherein changing the resonance condition polarizes each bath spin accordingly [8,9]. The same does not apply for identical spins characterized by a single frequency and a single resonance condition. Here collective dynamics and the conservation of angular momentum of the total bath-spin becomes the key physical picture [9]. Due to this, dynamics need to be solved within respective angular momentum subspaces that do not mix locking the polarization within individual subspaces. Henceforth, the bath does not reach its fully polarized state that is required for quantum sensing and computing protocols [1,2]. We have earlier shown that only by allowing the total-spin subspaces of the bath to mix, one can completely polarize it to its ground state [9].

Whereas cooling has been at the forefront for many quantum applications, controlled heating (mixing) has received less importance, as it is known that increasing entropy by some random irreversible process is easier than cooling, as it requires work to remove entropy [10]. However, for closed quantum systems, depositing (increasing) entropy is equally complex as cooling. This is due to intra-bath quantum correlations generated during the cooling and heating process as we show here. Establishing this symmetry and obtaining the rates for both processes for the same system has not been achieved. We will focus here on heating a quantum spin-bath that is initially in its highest polarized state. By coupling it to a two-level quantum probe that is periodically coupled to an infinite temperature bath, we ask how long does it take to empty its initial polarization (energy) and make it completely mixed.

Central spin-models can easily be realized in a variety of physical systems, of which the solid-state spin defect systems, quantum dots and NMR are at the forefront [1,5,11]. Central spin-based quantum batteries have been proposed and analyzed. These models have mainly focused on the charging of the batteries either unitarily [12] or dissipatively [13] or for interacting spin-baths [14]. We focus here on the discharging (depolarizing) spin batteries in a central-spin model and show how thermodynamical quantities such as energy and entropy get effected by the intra-bath correlations generated during the discharging process.

## 2. Materials and Methods

We consider an *N*-spin bath of identical spins, where each spin has an energy $\omega_{0}$ (ℏ=1) and is initially unpolarized. The Hamiltonian describing the bath
(1)$$H_{B}=\omega_{0}\sum_{k}S_{k}^{z},$$
where $S_{k}$ represents the spin-1/2 operator of any *k*th spin of the bath. Let us consider, for example, an initially unpolarized bath $\rho_{B}\left(0\right)=\frac{1}{2^{N}}I^{⊗N}$, where all the *N* spins are in their fully mixed state, i.e., $\rho_{B}^{k}=\frac{1}{2}I$, corresponding to the infinite temperature limit. The internal energy of such a bath state $E\equiv Tr[H_{B}\rho_{B}\left(0\right)]=0$. We now supply *N*-units of polarization, i.e., polarize each spin (pure state $\rho_{B}^{k}=|↑\rangle \langle ↑|$) such that the total energy of the spin-bath is $E=N\omega_{0}/2$. Given an ultra-long relaxation time for each bath-spin, this can act as an energy storage unit for polarization. The goal is now to extract the polarization/energy stored in the system. In doing so we would like to find the rates of decay of energy from the system and at the same time the rate of increase of entropy. We will show that while the total energy for the spin-bath goes to zero, the total entropy still does not reach its maximal limit, indicating the presence of non-extractable energy in the form of correlations.

The spin-bath is depolarized through a two-level quantum spin probe and is set in resonance with the spin-bath, i.e., $H_{P}=\omega_{0}\sigma^{z}$. The spin-probe is homogeneously coupled to bath-spins, allowing for identical rates in energy exchange with each bath-spin. We model this interaction through the well-studied central-spin model, wherein resonant energy exchange takes place between the probe and the bath. This is described by a simple flip-flop interaction given below,
(2)$$H=J\sum_{k}\left(\sigma^{+}S_{k}^{-}+\sigma^{-}S_{k}^{+}\right).$$

As the probe needs to extract energy, we initialize it in an unpolarized state, $\rho_{P}=\frac{1}{2}I$, by coupling it to an infinite temperature (hot) bath. Upon interaction, the probe transfers its entropy to the bath and extracts energy (polarization) from the bath. We again reset the probe to an unpolarized state by coupling it to the hot bath. We now repeat the process cyclically until no more polarization can be extracted from the bath. Upon reaching this asymptotic limit, we will evaluate both the net amount of energy extracted and the net amount of entropy deposited to the bath. Ideally, when the bath is completely depolarized (discharged), we expect that the bath is also in its highest entropy state the fully mixed state. We will show that this is not true given the continuous growth of correlations in the bath that allows for a finite ordering (symmetries) in the bath blocking the entropy to increase continuously. Only, upon breaking these symmetries, can one reset the bath to the hot-bath temperature.

The symmetries described above arise from the nature of interaction described in Equation (Equation 1). One can clearly see that the total spin angular momentum of the bath is conserved, i.e., $[H,\sum_{k}\sum_{k}S_{k}^{z}]=0$. Due to this symmetry, the Hamiltonian given in Equation (Equation 2) becomes block diagonal in the total-spin basis of the bath, and hence dynamics and energy exchange between the probe and the bath is restricted to a given total-spin sector (S) of the bath. Due to this, the above Hamiltonian can be simplified to
(3)$$H=J\sum_{S=0}^{N/2}H_{S}=J\sum_{S=0}^{N/2}\left[\sigma^{+}S^{-}+\sigma^{+}S^{+}\right]P_{S}.$$
where S=0…N/2 are the total spin sectors of the bath. The collective lowering and raising operators are given by $S^{\pm }=\sum_{k}S_{k}^{\pm }$. The projector $P_{S}$ ensures the block-diagonal evolution in each total spin sector *S* of the bath. For example, in the case of a two-spin bath, the total spin vectors are S=1,0, and one probe-bath interaction in Equation (Equation 3) will ensure that evolution happens independently in each spin sector of the bath, and no-mixing of these spin sectors will take place even in the presence of probe interaction.

The unitary evolution generated by the above Hamiltonian will also be block diagonal in the total-*S* basis of the bath, and hence can be simplified to
(4)$$U\left(t\right)=\Pi_{S}e^{iH_{S}t}P_{S}.$$

## 3. Results

Start from a fully polarized state of the bath, i.e., $\rho_{B}^{0}=|↑↑\ldots ↑\rangle \langle ↑↑\ldots ↑|=|N/2\rangle \langle N/2|$ that belongs to the total-spin sector $S=\frac{N}{2}$. The internal energy of this state as described earlier is $E^{i}=\frac{N\omega_{0}}{2}$. Using the depolarization scheme prescribed above, the bath state after M-interactions with the probe is modified to
(5)$$\rho_{B}^{M}=\sum_{m_{S}=-N/2}^{N/2}{C_{m}^{M}}_{S}|m_{S}\rangle \langle m_{S}|\approx \sum_{k=-N/2}^{N/2}e^{-\beta_{M}\omega_{0}m_{S}}|m_{S}\rangle \langle m_{S}|$$
where $\beta_{M}=1/k_{B}T_{eff}^{M}$, and $T_{eff}^{M}$ is the effective temperature that determines the population distribution of the bath state in the *M*th depolarization cycle. Such exponential relations hold for large *N*. We would like to note that assigning temperature for such far-from-equilibrium processes considered could be misleading. Such an assignment is made to show the heating of the bath state by interacting with a probe that is initially at infinite temperature. Such heating could also be evaluated through the entropy change, which we show in Figure 1d.

Taking into account the symmetry of the Hamiltonian, the energy exchange or the bath depolarization only happens within the spin-sector S=N/2, and hence in the asymptotic limit of M→∞, the steady state of the bath becomes
(6)$$\rho_{B}^{\infty }=\frac{1}{N+1}\sum_{m_{S}=-N/2}^{m_{S}=+N/2}|m_{S}\rangle \langle m_{S}|.$$Clearly, for the above state, $E^{\infty }\equiv Tr\left[H_{S}\rho_{B}^{\infty }\right]=0$, indicating that the bath is completely depolarized or, equivalently, the internal energy of each spin is zero. Upon tracing out the state of the *k*th bath spin, one finds that its state $\rho_{k}^{\infty }=\frac{1}{2}I$. On the other hand, the collective state of the bath $\rho_{S}^{\infty }$ is not the fully mixed state $\frac{1}{2^{N}}I^{⊗N}$. Henceforth, the entropy or mixedness of this asymptotic state is not maximal, and is given by
(7)$$P_{B}\equiv Tr\left[\rho_{B}^{\infty }\rho_{B}^{\infty }\right]=\frac{1}{N+1}.$$This is exponentially smaller than the purity of the expected end state, which is $1/2^{N}$.

Although the vanishing of the energy $E^{\infty }=0$ is itself not the measure to know whether the bath has been depolarized, as a unitary rotation of the initial pure state $\rho_{B}^{0}$ to a maximally entangled state (e.g., the GHZ state) can also have $E^{\infty }=0$ and the reduced state of each bath spin to be a fully mixed state. Henceforth, the depolarization or energy extraction can only be confirmed by evaluating the net energy increase of the probe in each cycle of interaction. To evaluate this, we will first focus on a simple test case of a two-spin bath, as the excitation transfer frequencies are commensurate, allowing for exact analytical calculation of the energy and entropy of the probe and the bath spins in each cycle.

### 3.1. Two-Spin Bath

Let us consider the case of a two-spin bath identically coupled to a spin-1/2 probe with coupling strength *J*. As there are two total-spin sectors S=1,0, and the zero-spin sector remains uncoupled, the dynamics effectively takes place in a single spin-sector S=1. The Hamiltonian given above simplifies to a two-body problem, and
(8)$$H=J(\sigma^{+}S^{-}+\sigma^{-}S^{+})$$
where $S^{\pm }$ are the raising and lowering operators of a spin-1 system. The fully polarized initial state of the bath $\rho_{B}^{0}=|↑↑\rangle \langle ↑↑|$ belongs to the total spin S=1 sector, making further analysis more straightforward. This state has two units of polarization, i.e., its energy $E_{B}\left(0\right)=2\omega_{0}$. We will start from the probe in a fully mixed state and repeatedly reset it back to the mixed state after a certain interaction time τ. After *M* such resettings, the two-spin state can be expressed as
(9)$$\rho_{B}\left(M\tau \right)=a_{M}\left(t\right)|1\rangle \langle 1|+b_{M}\left(t\right)|0\rangle \langle 0|+\left(1-a_{M}\left(t\right)-b_{M}\left(t\right)\right)|-1\rangle \langle -1|,$$
where |±1〉, |0〉 are the respective basis states of the spin-1 system. Ideally, we want coefficients to reach $a_{M}=b_{M}=1/3$. For an optimal interaction time τ=π/2J, the initial purity is lost (or the entropy is gained) at the rate
(10)$$P\left(M\right)\equiv Tr\left[\rho_{B}^{2}\right]=\frac{2+4^{M}}{3\cdot 4^{M}}=\frac{2}{3}e^{-M\log4}+\frac{1}{3},$$
reaching the asymptotic purity of 1/3, i.e., the bath is diagonal in the spin sector S=1. For this diagonal state, $E_{B}\left(m\tau \right)=0$, i.e., the two-spin state is depolarized. We need to check whether this lost polarization has been gained by the probe. For this we need evaluate the probe state after the *m*th cycle of interaction and prior to its reset. This is given by
(11)$$\rho_{P}\left(M\tau \right)=p_{M}\left(t\right)|↑\rangle \langle ↑|+\left(1-p_{M}\left(t\right)\right)|↓\rangle \langle ↓|$$
where $p_{M}\left(t\right)=\left(1+2^{M-1}\right)/2^{M}$, and the net energy gain by the probe
(12)$$E_{P}\left(M\tau \right)\equiv \omega_{0}\langle \sigma_{z}\rangle =\omega_{0}\sum_{M}\left(2p_{m}\left(t\right)-1\right)=2\omega_{0}=E_{B}\left(0\right).$$As expected, the probe gained (extracted) all the polarization from the two-spins, but the entropy has not yet reached its ideal value of 1/4. The asymptotic state of this depolarized state, which is comparatively less mixed than the fully mixed state is
(13)$$\rho_{B}\left(M\tau \right)=\frac{1}{3}|↑↑\rangle \langle ↑↑|+\frac{1}{3}|↓↓\rangle \langle ↓↓|+\frac{1}{6}\left[|↑↓+↓↑\rangle \langle ↑↓+↓↑|\right].$$For the above state $\langle S^{z}\rangle =0$, but higher order polarizations such as $\langle {\left(S^{l}\right)}^{2}\rangle =2/3$ (l=x,y,z), i.e., there is unextractable energy in the system, which is stored as correlations. This can also be verified by evaluating the quantum discord for the above state, which is
(14)$$Q(\rho_{B})=1/9.$$

Due to these quantum correlations, the entropy has not reached its maximal limit. To destroy these correlations, one needs to dephase the bath state and continue the depolarizing protocol to reach the fully mixed state that has maximal entropy. One may further wonder whether this form of correlation energy can be used to extract more energy than available initially. Using the normal dephasing control wherein the total angular momentum symmetry is broken and the two spin sectors S=1 and S=0 get mixed, one finds no change in the energy gained by the probe, but the entropy reaches its limiting value (1/4).

One can see that, while the energy extraction is already an exponential process ($E_{p}\left(t\right)=\omega_{0}\left(2-e^{-mln2}\right)$) the entropy enhancement (deposition) could be even longer due to the additional dephasing operations that are applied after the bath has reached its asymptotic state. The operations mix various spin sectors of the bath.

A similar analysis cannot be straightforwardly generalized for higher *N*, as the excitation transfer frequencies for N>2 are no more commensurate. Due to this, the closed form solutions for the bath state after the *m*th cycle is quite complex. We will rely on exact numerical diagonalization and random τ instead of a fixed optimal values used in the case of a two-spin bath.

### 3.2. *N*-Spin Bath

Due to incommensurable transfer frequencies for N>2, extending the above analysis analytically for arbitrary *N* is quite difficult. For this we rely on exact numerical diagonalization of the Hamiltonian and evaluate the changes in the net energy and entropy.

In Figure 2, we show the net gain in energy of the probe and the entropy of the bath with the cycle number *M*. We clearly see that the energy extraction rate only differs slightly with *N* and almost becomes independent with increasing *N*. The same trend is also seen in the purity (entropy) of the bath-state in Figure 3.

As described above, given that we can assign an effective temperature for the bath state for every depolarization cycle and obtain the heat, i.e., the net change in the internal energy and entropy of the state, it may appear that one could calculate the free energy. However, as this is a far-from-equilibrium situation and getting the exact value of $T_{eff}^{M}$ is difficult (given $\omega_{0}$ is variable), we do not estimate this quantity. Instead, from the change in internal energy $\Delta E_{M}=Tr\left(H_{B}\rho_{B}^{M}\right)-Tr\left(H_{B}\rho_{B}^{M-1}\right)$, and the change in entropy $\Delta S_{M}=S\left[\rho_{B}^{M}\right]-S\left[\rho_{B}^{M-1}\right]$, where $S\left[\rho \right]=-\sum_{m_{S}}{\lambda_{m}}_{S}\ln{\lambda_{m}}_{S}$ (see Figure 1), and given that the temperature in the *M*th cycle could be determined exactly, one could evaluate the change in the Gibbs free energy $\Delta G_{M}=\Delta E_{M}-T_{eff}\Delta S_{M}$.

### 3.3. Dephasing Control

As discussed earlier, although the net energy gain is independent of the correlations generated in the bath, the saturation value for the purity changes from 1/N+1 to $1/2^{N}$ as the correlations are gradually erased. We show in Figure 4 that such dephasing control is also beneficial for the probe, as the same energy can be extracted with a higher rate (power) when the bath is periodically dephased. Due to this, the bath spins will also reach their maximally mixed state, i.e., they are in equilibrium with the hot-bath similar to that of the probe.

## 4. Discussion

Extracting polarization (purity) from a quantum spin-bath is an exponential process similar to capacitor discharge in classical batteries. We also see here the role of many-body correlations, which mark a key role in differentiating thermalization (relaxation) and depolarization, i.e., we can identify a regime wherein entropy changes with no net energy changes, i.e., ΔE=0 before and after a depolarizing cycle. Correlations induced in the bath due to their common coupling with the probe are responsible for such entropy changes even though the energy has been extracted completely and no net energy changes occur in the system. These correlations also form a bottleneck in slowing down the energy extraction, and only upon erasing them periodically can one boost the power for depolarization (discharge), as shown in Figure 4.

These effects can be verified using solid state spin defects in diamond, where readily polarizable electron spins (NV centers) at room temperature are available [8]. For a typical 13C concentration (1 ppm), one can easily find 6 to 10 nuclear spins in the vicinity of the defect center with considerable coupling [4]. In the presence of the NV induced gradient, these nuclear spins become distinguishable, allowing one to polarize all the nuclear spins (spin-bath) to a high degree [4]. This will be the starting point for the protocol. The polarization within this nuclear ensemble can be stored for a long time (∼s), making it a very good energy storage unit. With this capability to polarize the central electron spin (the NV center) to any arbitrary value, and using long life time of the nuclear spins (>ms), the above protocol can be implemented. With an average coupling of 100 kHz, ∼10–50 depolarizing cycles can be implemented [15]. One can use half of the nuclear spins available for the depolarization protocol (discharging) and the other half to store the extracted polarization (charging). Further erasing correlations among the spins can be easily achieved using the NV-induced gradient field as shown in [9].

## 5. Conclusions

In conclusion, the findings here clearly identify the role of correlations in entropy and energy changes when depolarizing (extracting polarization from) a quantum spin-bath using a single quantum probe. Although the heat and entropy changes can be evaluated directly from the density matrix of the bath state, assigning temperature and its variation for the spin bath before and after interaction with the probe is not possible. This is due to the fact that we are dealing with a far-from-equilibrium process wherein finite size effects and short interaction time scales prohibit the complete thermodynamical description of the process. Due to this, we have focused here only on the thermodynamical quantities such as heat and entropy and how the correlations induced by probe–bath interaction effect them. We have also shown the possibility of defining effective temperature for the bath by approximately fitting the population distribution to the Boltzmann function. When using spin ensembles for realizing solid-state quantum batteries, it is essential to understand the role of correlations in both charging (polarizing) and discharging (depolarizing) of them. Although the stored energy can be completely extracted, the presence of correlations may effect their re-usage in battery applications and their efficiency [16,17].

## Figures and Tables

**Figure 1 entropy-25-00340-f001:**
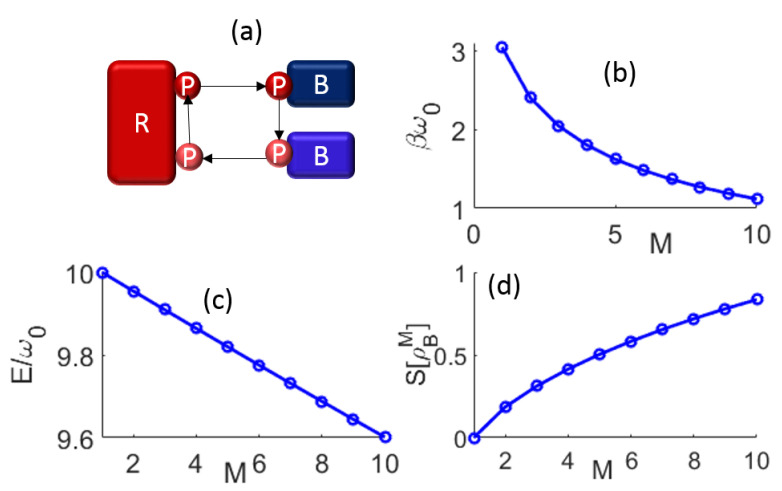
(**a**) The schematic representation of the depolarization scheme where a probe (P) is periodically reset by an infinite capacity hot bath reservoir (R) to a fully mixed state. The interaction of this high entropy probe with a low entropy spin-bath (B) results in entropy and energy exchange depicted through the change of color. (**b**) Increase in the effective temperature of the spin-bath after each depolarization cycle *M* is shown for an N=10 spin bath. For the same case, the (**c**) energy of the bath $E_{M}$ and (**d**) entropy of the bath state $S_{M}$ are plotted.

**Figure 2 entropy-25-00340-f002:**
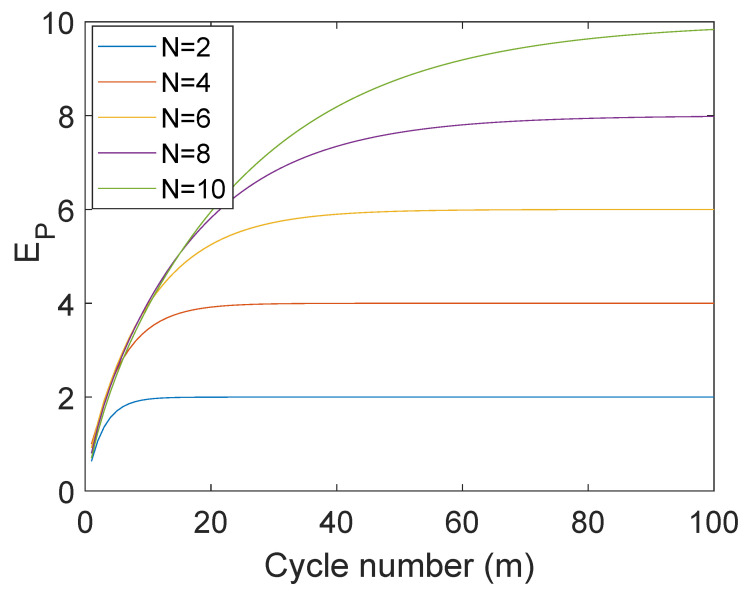
The net energy of the probe $E_{P}=\sum_{m}E_{P}\left(m\right)$ ($\hbar \omega_{0}=1$), i.e., the energy extracted from a *N*-spin bath is plotted as a function of depolarizing cycle *m*. The probe–bath interaction time-scale τ for each case is set to Jτ=π/2N.

**Figure 3 entropy-25-00340-f003:**
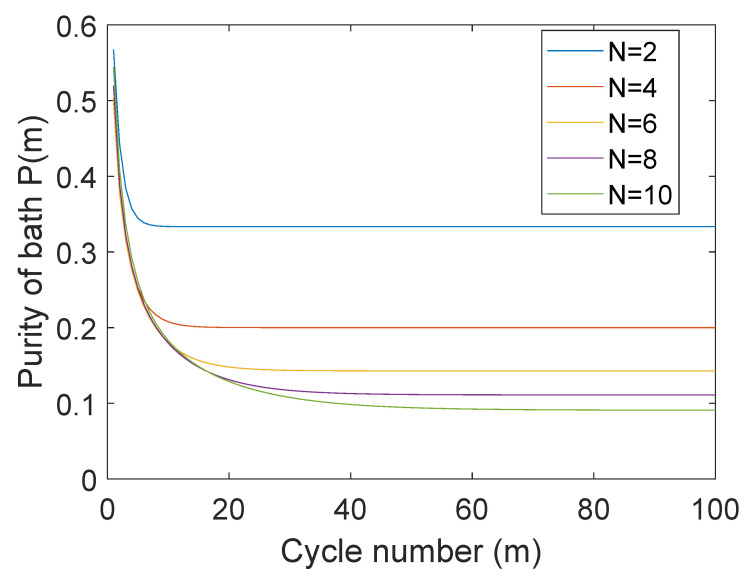
The purity of the *N*-spin bath after each depolarizing cycle *m* is plotted for various bath sizes *N*. The probe–bath interaction time-scale τ for each case is set to Jτ=π/2N.

**Figure 4 entropy-25-00340-f004:**
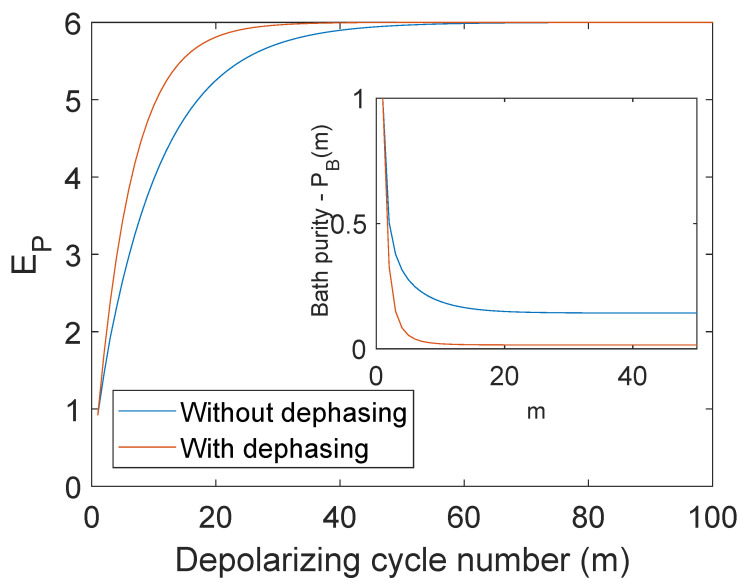
The net energy extracted by the probe $E_{P}$ ($\hbar \omega_{0}=1$) for the N=6 bath spins is plotted as a function of the depolarizing cycle *m*, for the cases with and without dephasing control on the bath.

## Data Availability

Not applicable.

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
