# Peer review of "Thermodynamics of Quantum Spin-Bath Depolarization"

_entropy, 2023, doi:10.3390/e25020340_

Round 1

Reviewer 1 Report

In the present paper the author investigate the dynamics of a quantum spin-bath

mainly focusing on its depolarization due to the coupling with a quantum probe. 

The work is interesting and can have some impact in the emerging field of quantum batteries in particular in view of recent experiments such as the one reported in Ref. [5] of the manuscript. 

According to this I recommend its publication on Entropy. 

I have few comments devoted to improve the quality of the work. They are listed in the following.

- The notation for the unpolarized state in line 64 is not clear to me. 

- Authors should spend more word to clarify the derivation of Eq. (3).

- I have the impression that some word is missing (for example in line 55 and line 81). The author should read carefully the manuscript. 

- The bibliography is quite limited, in particular concerning the quantum battery part, and slightly self-referential. This should be improved. 

Author Response

Dear Editor,

We are happy to receive the referee comments on our manuscript and have included their suggestions to clarify and improve the readability of the manuscript. We have added additional figure with data and schematics to clarify the main results of the paper. We have also made enough changes to text asked for and are resubmitting the modified version along with a point to point response to the referee’s comments.

We are hopeful that the changes made are satisfactory and are looking forward for its acceptance.

Thank you,

Best regards,

Durga Dasari

---------------------Referee 1----------------------------------------------

  1. In the present paper the author investigates the dynamics of a quantum spin-bath mainly focusing on its depolarization due to the coupling with a quantum probe. The work is interesting and can have some impact in the emerging field of quantum batteries in particular in view of recent experiments such as the one reported in Ref. [5] of the manuscript. According to this I recommend its publication on Entropy.

Response: We thank the referee for finding the manuscript interesting and beneficial for the emerging field of quantum batteries. We are also happy for his recommendation.

  1. The notation for the unpolarized state in line 64 is not clear to me.

Response: We thank the referee for the comment, we have added more text to clarify this.

  1. Authors should spend more word to clarify the derivation of Eq. (3).

Response: We have followed the suggestion of the referee and gave more details on Eq. (3)

  1. I have the impression that some word is missing (for example in line 55 and line 81). The author should read carefully the manuscript.

Response: We have carefully read the manuscript and made the editorial changes required.

  1. The bibliography is quite limited, in particular concerning the quantum battery part, and slightly self-referential. This should be improved.

Response: We have added two more relevant references to our work. We thank the referee for this useful comment.

Reviewer 2 Report

Referee report on paper: “Thermodynamics of Quantum spin-bath depolarization” by Durga Bhaktavatsala Rao Dasari

 In his paper, the author analyzes the thermodynamic effects in depolarizing a quantum spin-bath initially at zero temperature through a quantum probe coupled to an infinite temperature bath. He writes that because of quantum correlations the entropy has not reached its maximal limit.

 From my point of view, the paper makes a very confusion impression. The author writes that the spin-bath, being initially at zero temperature, contacts through a quantum probe with an infinite temperature bath.   In that case one could expect that the spin-bath becomes heated, thus has a finite temperature. Unfortunately, the author does not consider this case, and his description of the spin-bath does not take into account this fact. I suggest that the author has to consider the case of finite temperatures, including into consideration the free energy, E-TS. Without this step his research cannot be considered as a reliable. Also I suggest that the author would eliminate definitions like “Quantum spin-bath” written in capital letter. Such a definition leads to confusion while reading and understanding the paper.     

As I can understand and judge, the paper is very confusing and does not present a scientific articulated result.

 As a result, I suggest that the paper should be revised to clarify the above issues. In present form it is not suitable for publication.   

Author Response

Dear Editor,

We are happy to receive the referee comments on our manuscript and have included their suggestions to clarify and improve the readability of the manuscript. We have added additional figure with data and schematics to clarify the main results of the paper. We have also made enough changes to text asked for and are resubmitting the modified version along with a point to point response to the referee’s comments.

We are hopeful that the changes made are satisfactory and are looking forward for its acceptance.

Thank you,

Best regards,

Durga Dasari

--------------------------Referee 2-----------------------------------------

1. In his paper, the author analyzes the thermodynamic effects in depolarizing a quantum spin-bath initially at zero temperature through a quantum probe coupled to an infinite temperature bath. He writes that because of quantum correlations the entropy has not reached its maximal limit.

Response: We want to highlight both the extractable energy (heat) and the entropy change in depolarizing the bath. We agree with the referee that we need to give more details on the thermodynamic quantities to the reader to justify the case. Following his suggestion, we have now given the effective changes in the bath temperature and the entropy changes. We have a new figure Fig.1 for this.

2. From my point of view, the paper makes a very confusion impression. The author writes that the spin-bath, being initially at zero temperature, contacts through a quantum probe with an infinite temperature bath.   In that case one could expect that the spin-bath becomes heated, thus has a finite temperature.

Response: It is unfortunate that we have given a confusing impression to the referee about our manuscript. But we have clarified that the spin-bath would become mixed (heated) after every interaction slot with the probe. Though we have not defined a temperature change for this (difficult for such correlated cases), we have clearly shown the increase in mixedness (reduced purity) of the bath, as a measure for such heating of the bath.

In the modified version we have plotted the temperature of the bath depicting its heating by fitting the bath population approximately to Boltzmann distribution. Such temperature cannot be taken seriously as this is a far from equilibrium case with finite size bath effects. We have added text and equations clarifying the same. The results are shown in Fig.1.

3. Unfortunately, the author does not consider this case, and his description of the spin-bath does not take into account this fact. I suggest that the author has to consider the case of finite temperatures, including into consideration the free energy, E-TS. Without this step his research cannot be considered as a reliable. Also I suggest that the author would eliminate definitions like “Quantum spin-bath” written in capital letter. Such a definition leads to confusion while reading and understanding the paper.    

Response: We cannot completely agree with the referee on this comment. Calculating free energy in the form the referee suggested would require the knowledge of temperature, a quantity that is ill-defined in our case, which involves far from equilibrium scenario and a finite size bath. We have still followed referees suggestion and obtained the effective temperature. We give the expressions for heat and entropy, and wrote text on the possibility of evaluating the free energy when the temperature is exactly known. While a partial trace of each individual spin is a possible option, then there would be a huge discrepancy between the local and global temperatures.

4. As I can understand and judge, the paper is very confusing and does not present a scientific articulated result.

Response: We consider here an experimentally feasible scenario and the possible measurements that one can make in charging (polarizing) and discharging (depolarizing) large spin ensembles. While temperature for such far from equilibrium systems cannot be precisely determined, the possibility toe evaluate all Thermodynamic quantities would not be feasible. While the focus in only on the extraction of energy (polarization) and monitor the entropy changes and evaluate the role of correlations, we believe that enough justice has been made to articulate these results.

 5. As a result, I suggest that the paper should be revised to clarify the above issues. In present form it is not suitable for publication.  

Response: We are hopeful that the changes made are satisfactory and has improved the readability of the manuscript.

Round 2

Reviewer 2 Report

The author has corrected the text in the good direction. In my opinion, the paper can now be accepted for publication, provided that the author will add the next phrase to the Section 4, Conclusion; the phrase is taken from the author’s response.

 We have considered an experimentally feasible scenario and the possible measurements that one can make in charging (polarizing) and discharging (depolarizing) large spin ensembles. While temperature for such far from equilibrium systems cannot be precisely determined, the possibility to evaluate all thermodynamic quantities would not be feasible. We have focused in only on the extraction of energy (polarization), and evaluated the role of correlations. 

Author Response

Dear Editor,

We are happy to get the recommendation of both the referees. We have followed the referees suggestion and modified the conclusion accordingly. We are resubmitting the manuscript with these changes, and are looking forward for its acceptance.

best regards,

Durga Dasari

--------------------------Referee 2-----------------------------------------

The author has corrected the text in the good direction. In my opinion, the paper can now be accepted for publication, provided that the author will add the next phrase to the Section 4, Conclusion; the phrase is taken from the author’s response.

Response: We are happy that the refree finds the changes constructive and for recommending the publication. We have followed his suggestion and modified the conclusion to include the text from our previous response.